# Maghemite (γ-Fe_2_O_3_) and γ-Fe_2_O_3_-TiO_2_ Nanoparticles for Magnetic Hyperthermia Applications: Synthesis, Characterization and Heating Efficiency

**DOI:** 10.3390/ma14195691

**Published:** 2021-09-30

**Authors:** O. M. Lemine, Nawal Madkhali, Marzook Alshammari, Saja Algessair, Abbasher Gismelseed, Lassad El Mir, Moktar Hjiri, Ali A. Yousif, Kheireddine El-Boubbou

**Affiliations:** 1Department of Physics, College of Sciences, Imam Mohammad Ibn Saud Islamic University (IMISU), Riyadh 11623, Saudi Arabia; NAmadkhali@imamu.edu.sa (N.M.); saja.algessair@imamu.edu.sa (S.A.); 2The National Center for Laser and Optoelectronics, KACST, 6086, Riyadh 11442, Saudi Arabia; alshammari@kacst.edu.sa; 3Department of Physics, College of Science, Sultan Qaboos University, Code 123, Al Khoud P.O. Box 36, Oman; Abbasher@squ.edu.om (A.G.); ayousif@squ.edu.sa (A.A.Y.); 4Laboratory of Physics of Materials and Nanomaterials Applied at Environment (LaPhysMNE), Faculty of Sciences of Gabes, University of Gabes, Gabes 6072, Tunisia; elmirlassad@gmail.com (L.E.M.); M.Hjiri@kau.edu.sa (M.H.); 5Department of Physics, Faculty of Sciences, King Abdulaziz University, Jeddah 21589, Saudi Arabia; 6Department of Basic Sciences, College of Science & Health Professions, King Saud bin Abdulaziz University for Health Sciences (KSAU-HS), King Abdulaziz Medical City, National Guard Health Affairs, Riyadh 11481, Saudi Arabia; elboubboukh@ngha.med.sa; 7King Abdullah International Medical Research Center (KAIMRC), King Abdulaziz Medical City, National Guard Hospital, Riyadh 11426, Saudi Arabia

**Keywords:** iron oxide nanoparticles, maghemite, TiO_2_, Sol-Gel synthesis, magnetic hyperthermia, heating efficiency, alternating magnetic field

## Abstract

In this report, the heating efficiencies of γ-Fe_2_O_3_ and hybrid γ-Fe_2_O_3_-TiO_2_ nanoparticles NPs under an alternating magnetic field (AMF) have been investigated to evaluate their feasible use in magnetic hyperthermia. The NPs were synthesized by a modified sol-gel method and characterized by different techniques. X-ray diffraction (XRD), Mössbauer spectroscopy and electron microscopy analyses confirmed the maghemite (γ-Fe_2_O_3_) phase, crystallinity, good uniformity and 10 nm core sizes of the as-synthesized composites. SQUID hysteresis loops showed a non-negligible coercive field and remanence suggesting the ferromagnetic behavior of the particles. Heating efficiency measurements showed that both samples display high heating potentials and reached magnetic hyperthermia (42 °C) in relatively short times with shorter time (~3 min) observed for γ-Fe_2_O_3_ compared to γ-Fe_2_O_3_-TiO_2_. The specific absorption rate (SAR) values calculated for γ-Fe_2_O_3_ (up to 90 W/g) are higher than that for γ-Fe_2_O_3_-TiO_2_ (~40 W/g)_,_ confirming better heating efficiency for γ-Fe_2_O_3_ NPs. The intrinsic loss power (ILP) values of 1.57 nHm^2^/kg and 0.64 nHm^2^/kg obtained for both nanocomposites are in the range reported for commercial ferrofluids (0.2–3.1 nHm^2^/kg). Finally, the heating mechanism responsible for NP heat dissipation is explained concluding that both Neel and Brownian relaxations are contributing to heat production. Overall, the obtained high heating efficiencies suggest that the fabricated nanocomposites hold a great potential to be utilized in a wide spectrum of applications, particularly in magnetic photothermal hyperthermia treatments.

## 1. Introduction

The unique properties of magnetic iron oxide nanoparticles (NPs) confirmed its use in several applications such as photocatalysis, photonic, magnetic storage and electronic devices to biomedicine and theranostics [1,2]. Among these applications in clinical practice is their utilization in magnetic fluid hyperthermia (MFH) [3]. This is chiefly due to their excellent intrinsic magnetic properties, ultrasmall nanometer dimensions (5–15 nm) and their ability for dissipating heat using an alternating magnetic field (AMF). In particular, ferrites (mainly magnetite (Fe_3_O_4_) or maghemite (γ-Fe_2_O_3_)) are very promising materials for MFH and have been effectively utilized for cancer hyperthermia therapy [4,5,6]. Furthermore, a combination of iron-based nanoprobes with other transition metals (i.e., Ti, Au, Ag) at the nanoscale lead to formation of bifunctional materials which benefits from the unique properties of both components. Consequently, bifunctional iron oxide-based NPs, with both photo and magnetic properties, are expected to exhibit high potentials, particularly in multimodal photothermal therapies [7,8,9]. There is, therefore, a pressing need to understand the heat generated from such hybrid constructs and to show the influence of the added metal on the overall magnetism and heating properties of iron oxides.

The advantage of γ-Fe_2_O_3,_ the rare form of iron oxides, over other metal doped ferrite NPs is their magnetism and relatively high saturation magnetization. γ-Fe_2_O_3_ is a spinel ferrite and has almost the same structure as Fe_3_O_4_, but it converts to alpha hematite (α-Fe_2_O_3_) at very high temperatures [10,11,12]. Thus, the impeccable synthetic routes to achieve a series of uniform, size-controlled, highly-magnetic, crystalline and stable γ-Fe_2_O_3_ NPs remain unambiguously challenging. To date, several different methods have been employed to synthesize iron oxide NPs including non-aqueous and aqueous sol-gel, spray/laser pyrolysis, sonochemical, microemulsion, hydrothermal, chemical precipitation and thermal decomposition [13]. However, preparation of γ-Fe_2_O_3_ NPs continues to be difficult and often leads to phase transitions and loss of crystallinity during the synthesis [14]. Another important factor relies on employing modest, feasible and cost-effective route to prepare big quantities of NPs on demand. On the other hand, among other oxide semiconductors, TiO_2_ was the most interesting material due to its high photocatalytic activity, biochemical inertness, strong oxidizing power, relatively low price and long-term chemical and thermal stability against photo and chemical degradation. Therefore, the immobilization of TiO_2_ onto magnetic ferrite NPs could serve as an excellent material that could be used for wide range of applications (water treatment, disinfection, pollutant degradation and photothermal hyperthermia therapy). Most of the reports on iron oxide-TiO_2_ nanocomposites, however, focused their studies on the photocatalytic activities and showed that iron oxide NPs enhance the photocatalytic properties of TiO_2_ [15,16,17,18,19,20]. Only few research works investigated the heating efficiency of these nanocomposites for possible use in magnetic hyperthermia applications [19,20]. Shariful Islam et al. [19] studied the thermal-photocatalytic cell killing efficiency of Fe_3_O_4_–TiO_2_. It was found that the cancer cell killing percentage was enhanced by combining an altenating magnetic-field induction and UV–vis photoirradiation in comparison to only bare Fe_3_O_4_ or TiO_2_. Lu et al. [20] investigated the hyperthermia abilities and photocatalytic activities of porous γ-Fe_2_O_3_ microspheres decorated with TiO_2_. They reported a good heating ability of the composites, which make them promising candidate for magnetic hyperthermia applications.

In fact, magnetic hyperthermia using iron oxide NPs is currently boosting as such probes can act as efficient and local nanoheaters through remote activation (i.e., by AMF), which leads to remarkable therapeutic effects. However, their success relies on the precise control of their magnetic properties, sizes, specific absorption rate (SAR) and intrinsic loss power (ILP), as well as tuning parameters such as concentrations and amplitude of the applied magnetic field. The heat dissipated using an AC magnetic field is characteristically known by SAR, which is the amount of heat generated per unit gram of magnetic material and per unit time [21,22]. Previous studies reported that SAR values could be influenced by different parameters, among them, size, structure, magnetic properties, preparation methods, concentration and the frequency and amplitudes of the applied magnetic field [23,24,25,26,27]. De la Presa et al. prepared γ-Fe_2_O_3_ NPs by the technique called co-precipitation and demonstrated the effect of different parameters on SAR [24]. They found that the critical crystallite size to obtain maximum efficiency of heat was 12 nm. We previously investigated the heating efficiency of different sizes γ-Fe_2_O_3_ NPs using sol-gel technique [25,26] and we obtained that 14 nm was the best size to acquire preferable heating efficiency. In addition to the above parameters, the magnetic particle-particle interactions could also affect SAR values. Despite the large number of reports on the effect of variable parameters on SAR, the key factors affecting the heat dissipation is still not very well understood.

Herein, we report a simple process using modified sol-gel strategy to synthesize maghemite and maghemite-TiO_2_ nanocomposites. We tested whether the presence of TiO_2_ affect particle-particle magnetic interactions and induce any effects on heating abilities. We then studied the influence of magnetic properties and amplitude of the exercised magnetic field on heating abilities (SAR values) for the acquired particles. Finally, we discussed the plausible heating mechanism responsible for the generation of heat from the obtained NPs. The best we know, there is no report systematically studying different effects on the heating efficiencies of γ-Fe_2_O_3_ NPs and hybrid γ-Fe_2_O_3_-TiO_2_ NPs. These γ-Fe_2_O_3_@TiO_2_ nanocomposites, which integrate photo and magnetic properties, have great potential to be used in wide range of applications, particularly in medical photothermal hyperthermia.

## 2. Experimental

### 2.1. Synthesis of Maghemite and Maghemite-TiO_2_ Nanoparticles

In the first step, γ-Fe_2_O_3_ NPs were synthesized by a modified Sol-gel process in supercritical conditions of ethyl alcohol (EtOH) following similar procedure as in our previous work [25]. 5 g of iron (III) acetylacetonate [C_15_H_12_FeO_6_, 99%] obtained from Chemsavers was dissolved in a 30 mL of methanol (CH_3_OH, 98%) obtained from SIGMA-ALDRICH, under magnetic stirring of 400 rpm at room temperature for 15 min. The solution was then placed in an autoclave and dried under supercritical conditions of of ethanol (C_2_H_6_O, 96%) fabricated by Honeywell. The supercritical conditions of ethanol are P_c_ = 63.3 bar and Tc = 243 °C. The control of the heating of the autoclave was realised by temperature programmer.

In the second step, TiO_2_ NPs enriched by Fe_2_O_3_ particles were obtained by dissolving of 6.72 mmol of titanium (IV)-isoproxide (Ti(iOPr)_4_, 97%, from Chemsavers ) in a mixture of methanol and acetic acid (2 mL/2 mL). After 15 min of magnetic stirring, 50 mg of Fe_2_O_3_ NPs prepared in the first step was added to the solution and introduced in ultrasonic bath for 10 min. The temperature range was varied from ambient to 250 °C. The resulted solution was then introduced in an autoclave and dried in the supercritical condition of ethanol by using 600 mL.

This simple and cost-effective synthesis allowed the production of large quantities of γ-Fe_2_O_3_ NPs and γ-Fe_2_O_3_-TiO_2_ NPs on demand.

### 2.2. Structural Measurements

Using Bruker D8 Discover diffractometer (*θ*-2*θ*) equipped with Cu-Kα radiation (λ = 1.5406 Å), XRD analyses were made. The average crystallite size, D, of different samples was estimated by Scherrer’s formula [28]:(1)$$D=\frac{K\lambda }{\beta cos\theta }$$
where *θ* is the Bragg angle (in degree), *λ* is the incident wavelength (1.5406 Å), *K* is a constant whose value is approximately 0.9 and *β* (rad) is the full width at half maximum (FWHM) of a diffraction peak. Fourier transform infrared (FTIR) spectra were obtained using a Perkin-Elmer 580B IR spectrometer. The morphology of the samples was studied by means of transmission electron microscope (Type JEOL JSM-200F atomic resolution microscopy operating at 200 kV, Tokyo, Japan).

### 2.3. Magnetic Characterization

Mössbauer spectra were recorded at ambient temperature (RT = 295 K) and 78 K in standard transmission geometry using a constant acceleration signal spectrometer equipped with a ^57^Co source in a rhodium matrix. The data were analyzed using a non-linear least-squares fitting program assuming a Lorentzian distribution. Isomer shifts are presented in reference to α-Fe. Magnetic characterizations were performed in a Quantum Design MPMS-5S SQUID magnetometer (San Diego, CA, USA). Zero-field-cooled and field-cooled (ZFC-FC) curves were recorded at magnetic field of 100 Oe.

### 2.4. Heating Efficiency

A commercial system “Nanotherics Magnetherm” was used to carry out the heating efficiency of the samples under alternating current (AC) magnetic field as reported in our previous report [28]. The heat generated by magnetic nanoparticles for magnetic hyperthermia measurement is quantified by the *SAR*, which can be determined by:(2)$$SAR=\frac{\rho C_{w}}{Mass_{MNP}}\left(\frac{\Delta T}{\Delta t}\right)$$
where *C_w_* is defined as the specific heat capacity of water (4.185 J⋅kg^−1^⋅K^−1^), the density of the colloid is ρ,
the concentration of the magnetic nanoparticles in the suspension is called $Mass_{MNP}$ and the heating rate is represented by $\frac{\Delta T}{\Delta t}$. By performing a linear fit of temperature increase versus time at the initial time interval (1 to ~30 s), the slope ΔT/Δt is obtained. The influence of concentration (10 and 20 mg/mL) has been studied. To show the effect of different amplitudes (40, 90 and 130) of the magnetic field and 170 Oe at 332.8 kHz on the heating ability of the as-prepared NPs, one has selected a concentration of 10 mg/mL. All the samples are dispersed in deionized water

## 3. Results and Discussion

### 3.1. Structural Properties

XRD spectra of the obtained nanocomposites are shown in Figure 1. As can be seen in Figure 1a, XRD patterns of Fe_2_O_3_ NPs indicated the presence of diffraction peaks which correspond to spinel structure, which could be indexed on the basis of γ-Fe_2_O_3_ with space group P4132 (JCPDS No. 39-1346). No additional peaks have been observed suggesting that our synthetic method lead to the formation of a pure phase without any impurities that remain from the unreacted precursors. XRD patterns of γ-Fe_2_O_3_-TiO_2_ nanocomposite (Figure 1b) are similar to patterns obtained for γ-Fe_2_O_3_ but new peaks appear at 2*θ* = 25.3°, 37.86°, 48.2°, 54.1°, 55.2° are attributed to anatase-TiO_2_ [29], while the peaks located at 2θ = 27.57°, 41.02°, 54.3°, 68.96° are due to the rutile TiO_2_ [30]. We can conclude from XRD results the successful formation of γ-Fe_2_O_3_-TiO_2_ nanocomposite and that the addition of TiO_2_ did not induce any significant phase changes on the γ-Fe_2_O_3_ NPs. It is important also to note that the peaks for both samples are very broad indicating the formation of very fine particles. The average crystallite size obtained by using the Debye-Scherrer formula confirms this result. The average crystallite size estimated for γ-Fe_2_O_3_ was 8.5 nm. This value increased to 10 nm after adding TiO_2_ which has an average crystallite size of 16.5 nm and 22 nm for anatase and rutile phases, respectively.

FTIR was then extended to confirm the presence of Fe_2_O_3_ and Fe_2_O_3_-TiO_2_ (Figure 2). FTIR spectrum of γ-Fe_2_O_3_ showed a broad band at 3434 cm^−1^and 1630 cm^−1^ attributed to the O-H stretching and bending vibrations of surface hydroxyl, respectively. Another peak at 2927 cm^−1^ may corresponds to C-H groups stretching vibration. The band at 1420 cm^−1^ is attributed to the C-O stretching vibration, while the bands at 635 cm^−1^ and 583 cm^−1^ are associated with the Fe-O vibrational modes confirming the presence of iron oxide. The γ-Fe_2_O_3_-TiO_2_ shows a similar FTIR spectrum to that of γ-Fe_2_O_3_ with wide absorption band at 420–825 cm^−1^. This wide absorption band is attributed to Ti-O vibrations [20].

To better analyze the size and morphology of the obtained nanocomposites, transmission electron microscopy (TEM) was conducted. Figure 3a,b shows TEM images of γ-Fe_2_O_3_ and γ-Fe_2_O_3_-TiO_2_ nanocomposites, respectively. The images clearly indicate the quasi-cubic morphology of the samples with average core sizes of γ-Fe_2_O_3_ equal to 8.5 nm and slightly increases for γ-Fe_2_O_3_-TiO_2_ to 11 nm. The particle size distribution associated with the TEM images confirm the good uniformity of the as-synthesized NPs (Figure 3c,d).

### 3.2. Magnetic Characterization

#### 3.2.1. SQUID Measurements

Hysteresis loops and ZFC-FC curves for γ-Fe_2_O_3_ and γ-Fe_2_O_3_-TiO_2_ NPs are shown in Figure 4. The saturation magnetization (*Ms*), remanance (*Mr*), remnant to saturation magnetization ratio (*Mr*/*Ms*) and coercive field (*Hc*) values are presented in Table 1. It can be seen that the samples exhibit a non-negligible coercive field at room temperature, indicating that the particles do not behave as superparamagnetic (Figure 4a,b). The saturation magnetization at room temperature estimated for γ-Fe_2_O_3_ (*M*s = 84.5 emu/g) is higher than the standard value reported for bulk maghemite γ-Fe_2_O_3_ (*M*_s_ = 74 emu/g) but comparable to those reported in previous studies [24]. This is because of the spin disorder in surface that can be aligned readily in the applied magnetic field direction [31]. It might be also due to the presence of magnetite phase which has high magnetization, but XRD and Mossbauer results (discussed later) confirm the absence of this phase in the sample. The observed decrease of saturation of γ-Fe_2_O_3_-TiO_2_ NPs (*Ms* = 58.77 emu/g) is due to the non-magnetic nature of TiO_2_. Previous studies reported similar behavior of saturation after coating magnetite and maghemite with TiO_2_ [18,32]. To further understand the magnetic behavior of the samples, magnetization as a function of temperature using the zero-field-cooled and field-cooled (ZFC-FC) were performed (Figure 4c,d). The ZFC-FC curves indicate that blocking temperature (T_B_) for γ-Fe_2_O_3_ NPs is above room temperature, while T_B_ is around 240K for γ-Fe_2_O_3_-TiO_2_ NPs.

The coercive field H_C_ increases with decreasing temperature for both samples due perhaps to the blocking of magnetic moments. The behavior of H_C_ at room temperature for the two samples can be analyzed in term of size. It can be seen that coercivity at room temperature increase slightly for γ-Fe_2_O_3_-TiO_2_ nanocomposite (*H_C_* = 27 Oe) compared to γ-Fe_2_O_3_ (*H_C_* = 23 Oe). This behavior can be understood on the basis of model describing the magnetic behavior in the monodomain regime; where coercivity follow the relation [24,33]:(3)$$H_{C}=\left(\frac{2K_{eff}}{\mu_{0}M_{s}}\right)\left[1-\frac{25K_{B}T}{K_{eff}V}\right]$$
where $K_{eff}$ is effective anisotropy constant, *M_S_* is the saturation and *V* is the volume of the nanoparticles. It can be seen from the relation (3) that the coercive field depends on saturation, the volume and effective anisotropy constant. All these parameters are size dependent and that can explain the decrease of the coercivity with decreasing size.

Using the law of approach to saturation (LAS), we attempt to calculate the anisotropy constant $K_{eff}$ which characterize the magnetization close to the saturation as below [34,35]:(4)$$M\left(H\right)=M_{s}\left(1-\frac{b}{H^{2}}\right)$$
where *b* is a parameter used to determine *K_eff_* by using the following equation [25]:(5)$$K_{eff}=\mu_{0}M_{s}\sqrt{\frac{15b}{4}}$$

The calculated value of $K_{eff}$ for both samples at 10 K and 300 K are summarized in Table 1. As can be seen, the effective anisotropy constant obtained for γ-Fe_2_O_3_ NPs at room temperature ($K_{eff}$ = 5.68 × 10^4^ erg/cm^3^) is close to that reported for bulk γ-Fe_2_O_3_ ($K_{eff}$ = 4.7 × 10^4^ erg/cm^3^) [36].

#### 3.2.2. Mössbauer Spectroscopy

We indexed the phase of the obtained NPs as γ-Fe_2_O_3_ from XRD, but it could be also indexed as Fe_3_O_4_ given that XRD patterns of γ-Fe_2_O_3_ and Fe_3_O_4_ are almost the same. In order to differentiate between the two phases (γ-Fe_2_O_3_ and Fe_3_O_4_) and to confirm that the obtained phase is γ-Fe_2_O_3_, Mössbauer spectroscopy studies were carried out. It is well known that the hyperfine parameters such as isomeric shift and hyperfine field of both Fe-oxides are different. The Mössbauer spectra of the two samples at 78 K and 295 K are shown in Figure 5a and as can be observed, all the spectra show well-defined magnetic sextet pattern. The best fit was obtained by considering two dominating sextets and one small magnetic component A, B and C, respectively, as shown in Figure 5b. As can be seen from Table 2, the isomeric shift values of the main components A (0.40 mm/s) and B (0.43 mm/s) of Fe_2_O_3_ at 78 K are typical of ferric (Fe^3+^) iron [37]. In addition to the isomer shift values, the hyperfine field and quadrupole shifts of both components are the typical characteristics of Fe^3+^ ions in maghemite γ-Fe_2_O_3_ nanoparticles [38]. Furthermore, the hyperfine parameters of the third component C is also attributed to the Fe^3+^ ions. It is important to highlight the absence of ferrous (Fe^+2^), which is characteristic of magnetite (Fe_3_O_4_). Thus, it can be concluded that the synthesized NPs are indexed as γ-Fe_2_O_3_ with space group P4132 (JCPDS No. 39-1346). The A and B components are attributed to the tetrahedral and octahedral sites of maghemite, respectively, while the third component C is attributed to the iron ions in the surface layer. The spectrum of Fe_2_O_3_-TiO_2_ at 78 K was fitted with same components and no changes are observed except slight decreases in the hyperfine field values due to the effect of TiO_2_, which is non-magnetic. This is in agreement with magnetization measurement that clearly shows a decrease of Fe_2_O_3_-TiO_2_ saturation.

The Mössbauer spectra for both samples show well-defined magnetic sextet pattern, but the lines become broadened (Figure 5), indicating that the samples conserved their magnetic order remain. This is in agreement with SQUID measurements, which indicated that both samples are ferromagnetic at room temperature. The phase is confirmed again by fitting the spectra with the same components corresponding to Fe^3+^ ions in the tetrahedral, octahedral and in the surface layer. For γ-Fe_2_O_3_-TiO_2_ sample, the best fit was obtained by adding paramagnetic doublet component D (Table 2), which is attributed to the transition of some part of NPs from magnetic ordering to a paramagnetic state (Figure 6b). This paramagnetic component is due to part of the NPs, where their blocking temperature T_B_ is below 295 K. ZFC/FC measurements corroborated this hypothesis and showed that γ-Fe_2_O_3_-TiO_2_ sample has T_B_ around 240 K, while T_B_ for γ-Fe_2_O_3_ is above room temperature. In summary, the Mössbauer results indicated that the phase is maghemite and that both samples are ferrimagnetic at room temperature, which is in agreement with magnetic measurements.

### 3.3. Heating Efficiency Measurements

#### 3.3.1. SAR as Function of Concentration

The heating efficiencies of γ-Fe_2_O_3_ and γ-Fe_2_O_3_-TiO_2_ nanocomposites dispersed in deionized water at different concentrations under AMF with frequency and amplitudes that satisfy the magnetic hyperthermia safety condition (Hxf ≤ 5 × 10^9^A/m.s) [24] were conducted (Figure 6). The main heating parameters obtained from the temperature rise are summarized in Table 3. As can be seen in Figure 6a,b, all the samples show high heating ability and reach magnetic hyperthermia temperature (42 °C) in short time. For instance, γ-Fe_2_O_3_ reached magnetic hyperthermia temperature in only 3 min, while γ-Fe_2_O_3_-TiO_2_ took around 4.5 min to reach the same temperature at the same concentration. As expected, the rise in temperature decreases with decreasing the concentration for both samples. While 20 mg/mL sample of γ-Fe_2_O_3_ could reach high temperatures up to 73 °C in 15 min, temperature up to 62 °C was achieved for 10 mg/mL concentration of the same sample. This increase can be explained by the additional amount of magnetic nanostructure, which is generally the main source of heat dissipation. The calculated values of SAR as function of concentration are shown in Figure 6c. It can be seen that SAR values of γ-Fe_2_O_3_ are higher than that obtained for γ-Fe_2_O_3_-TiO_2_ nanocomposites but both samples have considerable SAR values. As reported by previous reports, interparticles dipolar interaction increases with increasing concentration of NPs and that could induce such effect on the heating [22,39,40]. Abbasi et al. [39] claimed that increase of dipolar interaction with enhancing concentration has a considerable influence on the Neel relaxation time (which will be discussed later).

#### 3.3.2. SAR as Function of Field Amplitude

Figure 7a,b shows the temperature rise of the samples with concentrations of 10 mg/mL in frequency of 332.8 kHz and for different strength of AC magnetic field. When the field amplitude increases from 90 Oe to 170 Oe and as expected, an increase of the temperature is observed (Figure 7c), indicating that the heating efficiency of the NPs can be tuned by changing the field amplitude. As can be observed also in Figure 8a, SAR increases with increasing field amplitude and reached their higher values at field amplitude of 170 Oe for both samples. The same trend of SAR with the field amplitude was reported for many NP systems [22,24,25]. Furthermore, we investigated that the linear response theory (LRT) was valid for the two samples. In this model, *SAR* varies linearly as a function of square of field amplitude and given as below [24]:(6)$$SAR=cfH^{2}$$
where *c* is a constant, f is the frequency and *H* is amplitude of the field.

Figure 8b shows that field amplitude dependence of experimental SARs has a quadratic behavior as expected by the LRT model. The coefficient of determination R^2^, which should be near 1 for better fit is around 0.998 for both samples and this affirms the high fit accuracy.

#### 3.3.3. Comparison of Heating Ability

Both samples reached magnetic hyperthermia (42 °C) in a relatively short time but the time needed to reach this temperature is shorter in the case of γ-Fe_2_O_3_ compared to γ-Fe_2_O_3_-TiO_2_ nanocomposites. Furthermore, the SAR values achieved for γ-Fe_2_O_3_ are higher than that obtained for γ-Fe_2_O_3_-TiO_2,_ indicating better heating for γ-Fe_2_O_3_ NPs. This high heating efficiency could be explained mainly by the high saturation of γ-Fe_2_O_3_ NPs (*Ms* = 84.55 emu/g) compared to 58.77 emu/g obtained for γ-Fe_2_O_3_-TiO_2_. Other parameters such as size of NPs and the effective anisotropy constant (*K_eff_*) can also affect SAR. However, the effect of size can be neglected due to almost comparable sizes obtained from TEM measurements, while *K_eff_* of γ-Fe_2_O_3_ (5.68 × 10^4^ erg/cm^3^) is about 16-fold larger than γ-Fe_2_O_3_-TiO_2_ nanocomposite value (3.43 × 10^3^ erg/cm^3^).

The comparison of the heating ability of our NPs with other systems through the SAR values is depicted in Table 4. However, this comparison of SAR values does not give much information on the heating efficiency given that each study has its own experimental conditions such as concentration, magnetic properties, field amplitude, frequency etc.

To allow a logic comparison of the heating efficiency, we used the intrinsic loss power (ILP), which is given by [38]:(7)$$ILP=SAR/fH_{0}^{2}$$
where *f* is the frequency and *H*_0_ is magnetic field.

It can be seen from Table 3 that ILP values for 10 mg/mL sample of γ-Fe_2_O_3_ (1.57 nHm^2^/kg) is larger than that achieved for the same concentration of γ-Fe_2_O_3_-TiO_2_ (0.64 nHm^2^/kg) showing again the high heating efficiency of γ-Fe_2_O_3_ compared to γ-Fe_2_O_3_-TiO_2_ nanocomposites. However, the ILP values for both samples are in the range reported for commercial ferrofluids (0.2–3.1 nHm^2^/kg) [42].

#### 3.3.4. Mechanism of Heating

Heat dissipation from magnetic nanoparticles under AC magnetic field is caused by three major mechanisms, namely: Hysteresis loss; Brownian relaxation and Néel relaxation as discussed in our previous report [22,25]. We and others [41,43] believe that all the three mechanisms are collaboratively effective in the heat generation. However, some aspects are more dominant over the others as discussed below.

Generally, hysteresis losses are in proportion with the area (A) of the hysteresis loop. As can be observed from M-H curves (Figure 4), both samples present a minor hysteresis loop area due to the low values of coercivity and remenance. It is therefore reasonable to deduce that the contribution of hysteresis loss in the heat dissipated by the samples can be neglected.

In Néel relaxation, the magnetic moment of NPs suspended in fluid can relax after magnetic field removal and relaxation time is given by:(8)$$\tau_{N}=\tau_{0}e^{K_{eff}V/k_{B}T}$$
where *τ*_0_ = 10^−9^ s, V the particle volume, *K_eff_* magnetic anisotropy constant, *k_B_* the Boltzmann constant and *T* the absolute temperature.

The entire nanoparticles can rotate through Brownian relaxation during time *τ_B_*:(9)$$\tau_{B}=\frac{3\eta V_{h}}{k_{B}T}$$
where *η* is the viscosity of the fluid and *V_h_* the particle volume.

It can be seen that Néel relaxation time depends exponentially on volume and magnetic anisotropy whereas the Brownian relaxation varies linearly with the volume and the media viscosity.

In most cases, the combination of the two mechanisms is more suitable, but the quick relaxation mechanism is prevalent and an effective relaxation time may be known as:(10)$$\frac{1}{\tau_{eff}}=\frac{1}{\tau_{N}}+\frac{1}{\tau_{B}}$$

Previous studies showed that Néel relaxation is more dominating in the case of particles with smaller sizes, while larger particles relax in liquid medium mainly by a Brownian mechanism [24,25,43,44]. The NP sizes deduced from XRD and TEM for our samples allow us to suppose that the contribution of Néel relaxation is more dominant than that of Brownian relaxation. In order to confirm the contribution of Brownian relaxation in the heat dissipated by NPs, the temperature rise was measured in different carrier liquids as shown in Figure 9. It is expected that Brownian relaxation time in liquid with lower viscosity will decreases as described by Equation (9) and that would induce an increase of heating efficiency. It can be observed from Figure 9, that the sample dispersed in acetone reach magnetic hyperthermia (42 °C), while sample in deionized water does not reach this temperature. The heating efficiency clearly increase for NPs dispersed in acetone which has lower viscosity (0.295 mPa.s) compared to deionized water (0.89 mPa.s). Thus, we can conclude that Brownian relaxation mechanism is also contributing to the heat production.

## 4. Conclusions

In conclusion, a modified sol-gel method was employed to synthesize γ-Fe_2_O_3_ and γ-Fe_2_O_3_-TiO_2_ nanocomposites with small sizes and good uniformity for magnetic hyperthermia applications. Magnetization measurements and heating efficiencies were investigated in detail. The influence of concentration, magnetic field amplitude and carrier liquid on heating efficiency was presented. Our results show that the ILP and SAR values differ slightly between the two samples, but both have high heating efficiency and reach magnetic hyperthermia temperature (42 °C) in relatively short times. While γ-Fe_2_O_3_ NPs reached magnetic hyperthermia temperatures in 3 min, γ-Fe_2_O_3_-TiO_2_ NPs require around 10 min for reaching the same temperature. SAR values indicated that the heating efficiency of the NPs can be tuned by changing the field amplitude or concentration of the NPs. The dependence of SAR values with field amplitude follows linear response theory (LRT). Moreover, the ILP values of 1.57 nHm^2^/kg and 0.64 nHm^2^/kg obtained for γ-Fe_2_O_3_ and γ-Fe_2_O_3_-TiO_2_, respectively, are in the range reported for commercial ferrofluids (0.2−3.1 nHm^2^/kg), showing their good heating efficiency. The high crystallinity, good SAR and ILP values make these NPs promising candidates for hyperthermia application.

## Figures and Tables

**Figure 1 materials-14-05691-f001:**
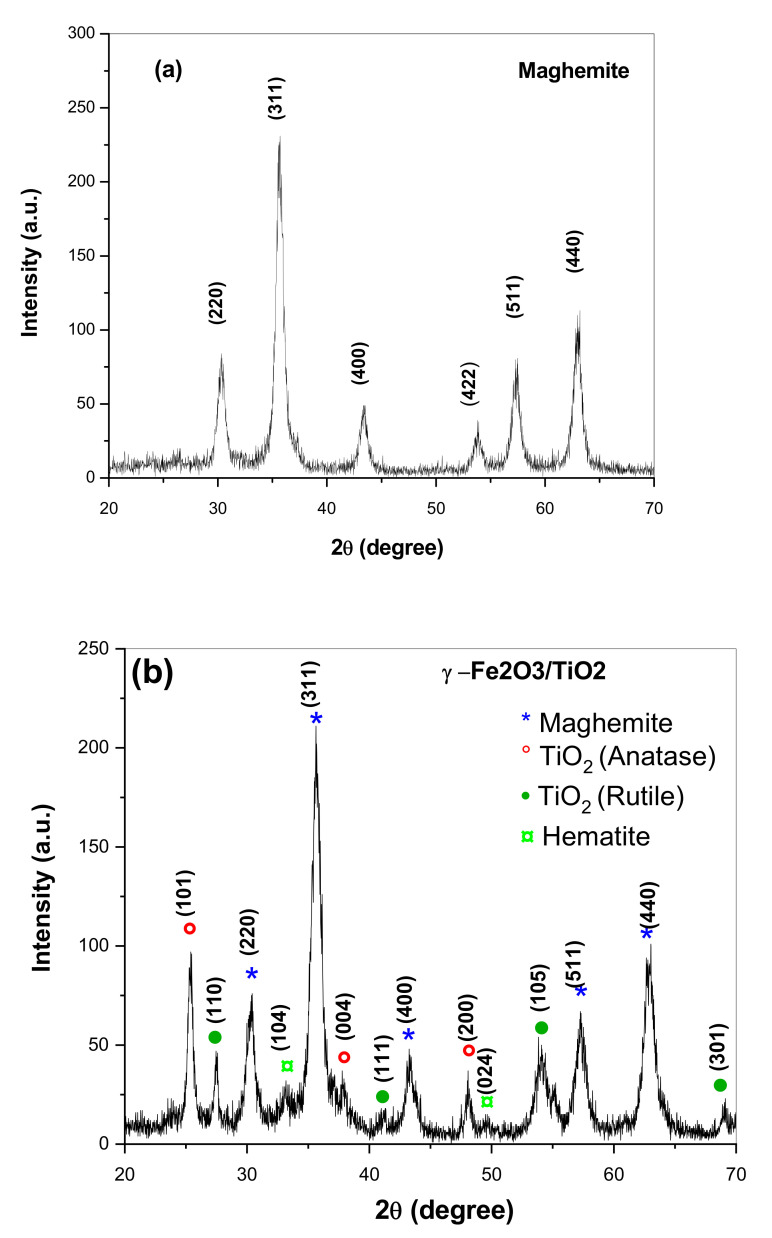
XRD patterns of (**a**) γ-Fe_2_O_3_ and (**b**) γ-Fe_2_O_3_-TiO_2_ nanocomposites.

**Figure 2 materials-14-05691-f002:**
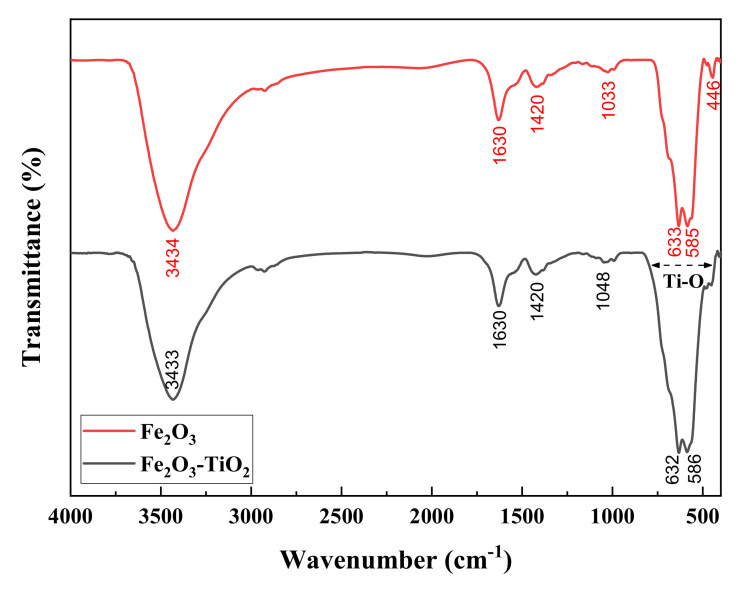
FTIR spectra of γ-Fe_2_O_3_ (up) and γ-Fe_2_O_3_-TiO_2_ (down) NPs.

**Figure 3 materials-14-05691-f003:**
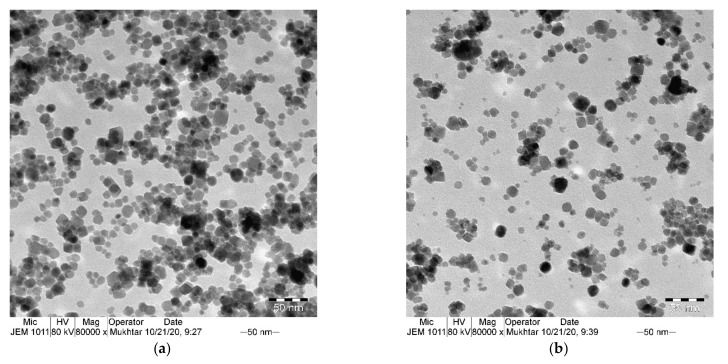
TEM photographs (**a**,**b**) particles size distribution (**c**,**d**) γ-Fe_2_O_3_ and γ-Fe_2_O_3_-TiO_2_ nanocomposite respectively.

**Figure 4 materials-14-05691-f004:**
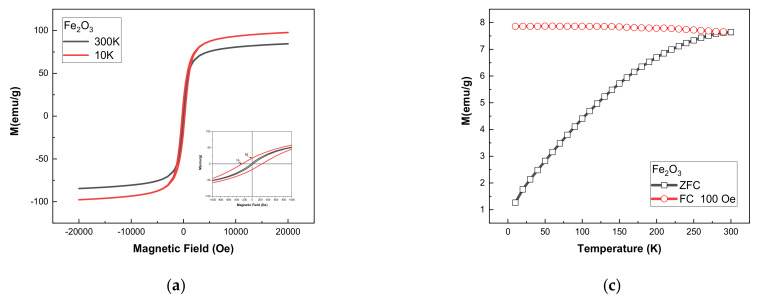
Hysteresis loops of (**a**) γ-Fe_2_O_3_ and (**b**) γ-Fe_2_O_3_-TiO_2_ nanocomposites. ZFC/FC curves of (**c**) γ-Fe_2_O_3_ and (**d**) γ-Fe_2_O_3_-TiO_2_.

**Figure 5 materials-14-05691-f005:**
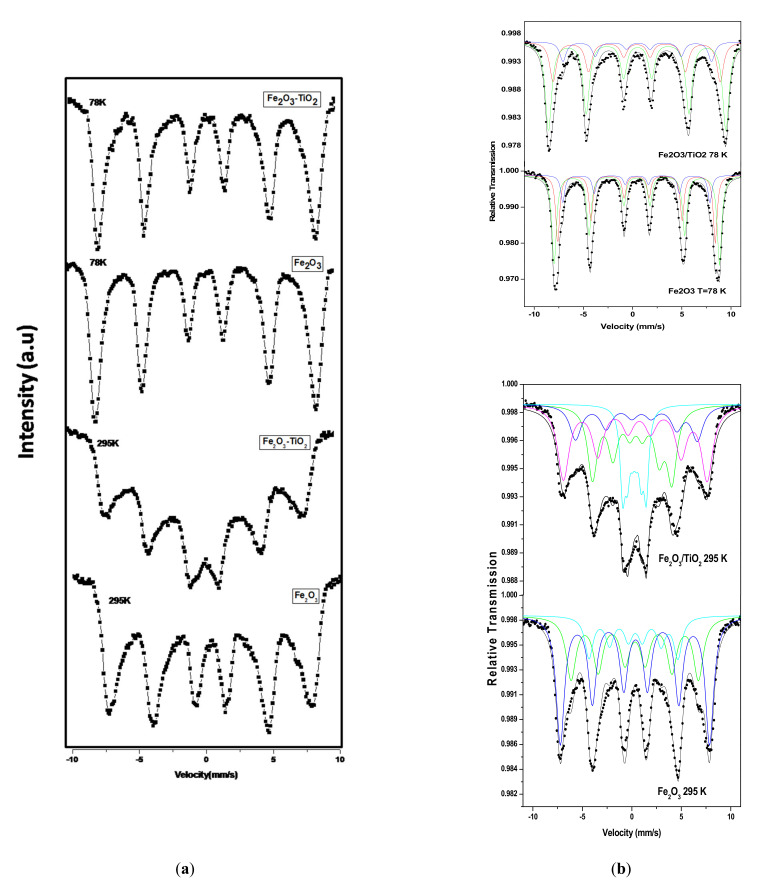
(**a**) The ^57^Fe Mössbauer spectra of γ-Fe_2_O_3_ and γ-Fe_2_O_3_-TiO_2_ nanocomposite at 78 and 295 K; (**b**) The calculated spectrum components for both samples corresponding to the tetrahedral (A), octahedral (B) sites of Fe^3+^ ions and iron ions at the surface of the NPs (C).

**Figure 6 materials-14-05691-f006:**
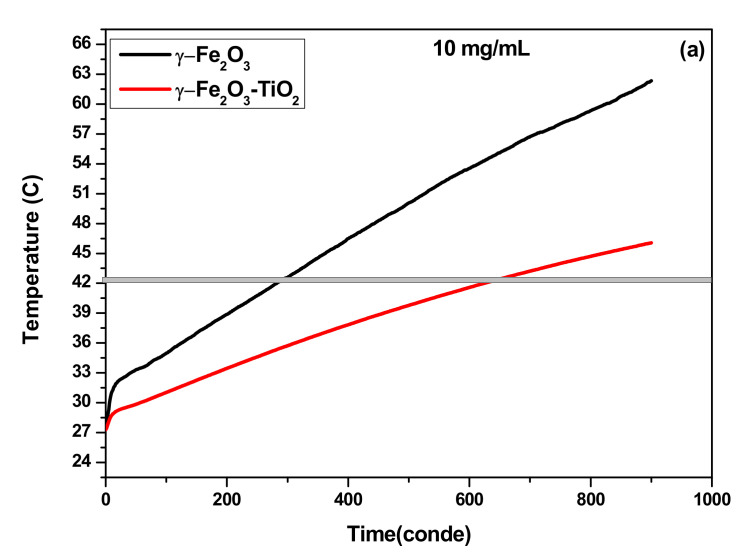
Temperature increases at H_0_ = 170 Oe and *f* = 332.8 kHz for γ-Fe_2_O_3_ and (**b**) γ-Fe_2_O_3_-TiO_2_ nanocomposite: (**a**) 10 mg/mL, (**b**) 20 mg/mL and (**c**) SAR values as function of concentration.

**Figure 7 materials-14-05691-f007:**
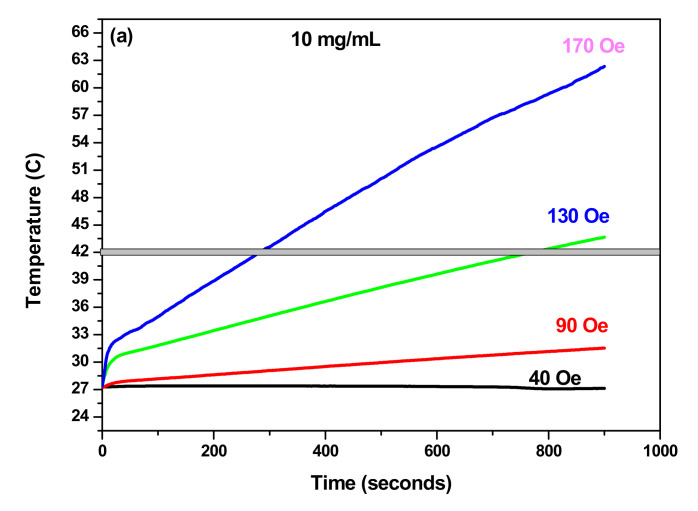
Temperature increases at *f* = 332.8 kHz and different AC magnetic field of (**a**) γ-Fe_2_O_3_, (**b**) γ-Fe_2_O_3_-TiO_2_ and (**c**) Temperature *vs* field amplitude.

**Figure 8 materials-14-05691-f008:**
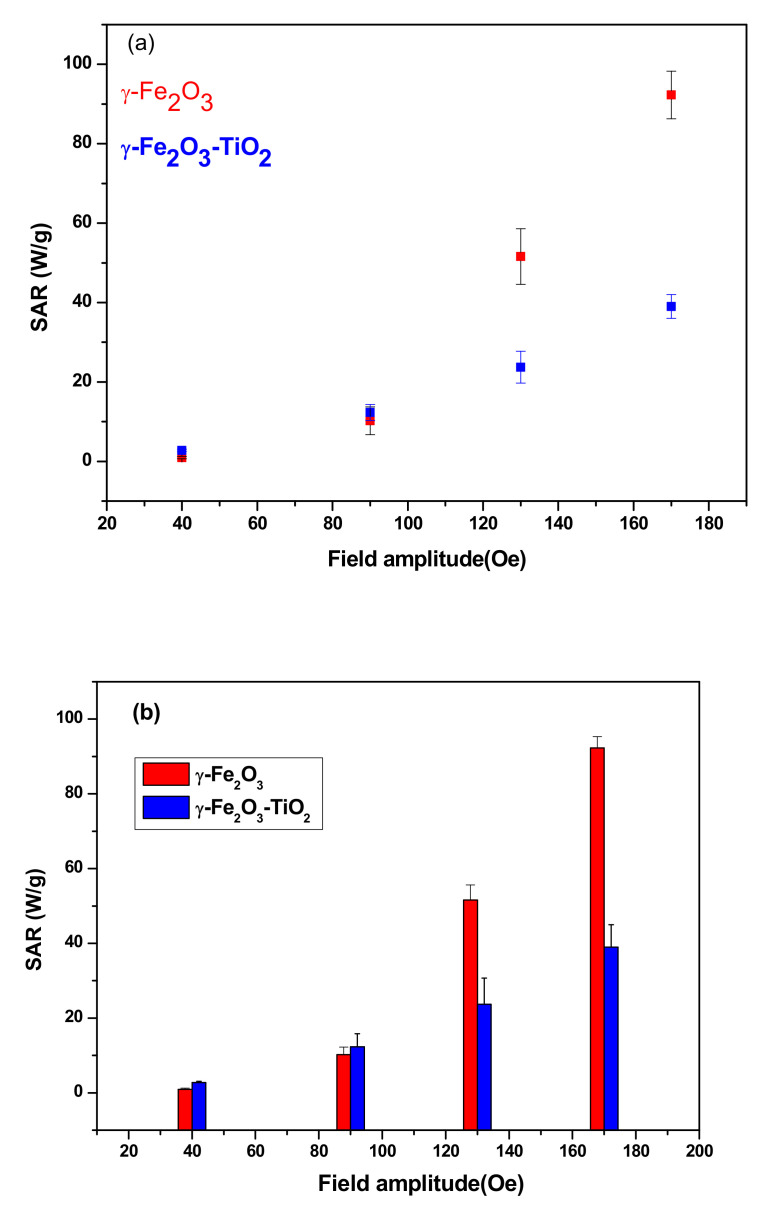
(**a**) SAR vs. field amplitude for both samples and (**b**) LRT model showing the evolution of SAR with square of field amplitude.

**Figure 9 materials-14-05691-f009:**
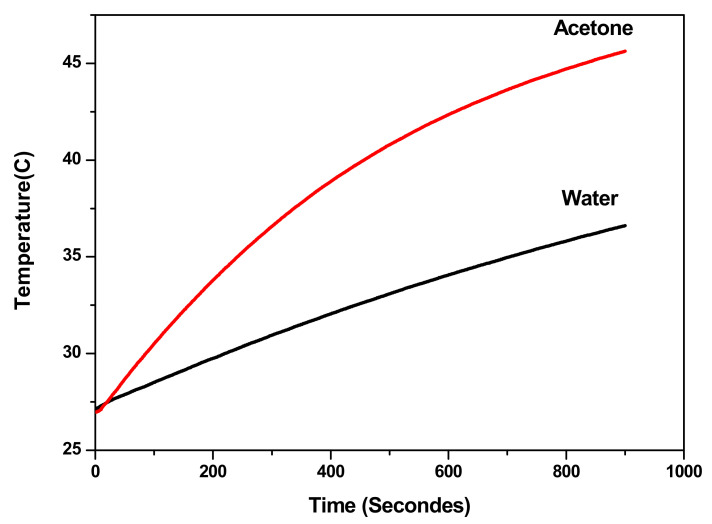
Increase in temperature at *H*_0_ = 170 Oe and *f* = 332.8 kHz for γ-Fe_2_O_3_-TiO_2_ NPs (5 mg/mL) dispersed in deionized water and in acetone.

**Table 1 materials-14-05691-t001:** Magnetic parameters deduced from hysteresis loops and anisotropy constant (*K_eff_*) for γ-Fe_2_O_3_ and γ-Fe_2_O_3_-TiO_2_.

Samples	*M_s_* (emu/g)		*M_r_* (emu/g)		*M_r_*/*M_s_*		*H_c_* (Oe)		*K_eff_* (erg/cm^3^)		D (nm)
	10 K	300 K	10 K	300 K	10 K	300 K	10 K	300 K	10 K	300 K	
Fe_2_O_3_	97.76	84.55	17.51	2.69	0.179	0.032	252.17	23.26	6.44 × 10^3^	5.68 × 10^4^	8.5
Fe_2_O_3_-TiO_2_	68.11	58.77	12.25	3.11	0.18	0.053	217.51	27	3.51 × 10^3^	3.43 × 10^3^	11

**Table 2 materials-14-05691-t002:** Hyperfine parameters deuced from the Mossbauer spectra at 78 and 295 K.

**Samples**	**Component**	**Isomer Shifts δ ± (0.01) mm/s**		**Quadrupole Shifts ε ± (0.002) mm/s**		**Magnetic Hyperfine Field** **B_hf_ (±0.1)** **T**		**Area** **A (±1) %**	
		78 K	295 K	78 K	295 K	78 K	295 K	78 K	295 K
Fe_2_O_3_	A	0.40	0.31	−0.007	−0.021	49.7	40.0	34	33
	B	0.43	0.34	−0.004	−0.029	52.1	46.7	52	51
	C	0.40	0.26	−0.024	−0.078	46.4	27.8	14	16
Fe_2_O_3_/TiO_2_	A	0.37	0.70	−0.013	−0.253	47.9	38.2	32	17
	B	0.43	0.54	−0.020	−0.211	50.9	45.1	55	36
	C	0.46	0.21	−0.045	−0.191	42.6	24.9	13	30
	D	−	0.24	−	−0.009	−	7.4	−	17

**Table 3 materials-14-05691-t003:** Heating characteristics at H_0_ = 170Oe and f = 332.8 kHz.

ILP (nH m^2^/kg)	SAR (W/g)	Time Needed to Reach Hyperthermia Temperature 42 °C (min)	Maximum Temperature (°C)	Concentration	Samples
1.57	92.3	4.8	62	10 mg/mL	γ-Fe_2_O_3_
0.64	39	10.5	46		γ-Fe_2_O_3_-TiO_2_
0.73	44.46	3	73	20 mg/mL	γ-Fe_2_O_3_
0.67	41.15	4.6	57.5		γ-Fe_2_O_3_-TiO_2_

**Table 4 materials-14-05691-t004:** Comparison of ILP values for different magnetic NPs.

Magnetic Nanoparticles	Synthesis Method	Frequency (kHz)	Field (Oe)	Medium	ILP (nHm^2^/kg)	Ref
γ-Fe_2_O_3_	Modified Sol-gel	332.8	170	distilled water	1.57	This work
γ-Fe_2_O_3_	Modified Sol-gel	523	100	distilled water	1.3	[25]
γ-Fe_2_O_3_ -TiO_2_	Modified Sol-gel	332.8	170	distilled water	0.64	This work
γ-Fe_2_O_3_@TiO_2_	Hydrothermal	55	86.7	Physiological saline	−	[20]
Fe_2_O_3_	Hydrothermal	200	251.3	distilled water	1.12	[40]
Zn _0.1_Fe_0.9_Fe_2_O_4_	coprecipitation method	339	92	distilled water	5.4	[41]

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
