# Peer review of "Maghemite (γ-Fe2O3) and γ-Fe2O3-TiO2 Nanoparticles for Magnetic Hyperthermia Applications: Synthesis, Characterization and Heating Efficiency"

_materials, 2021, doi:10.3390/ma14195691_

Round 1

Reviewer 1 Report

This manuscript has some mistakes in the English writing and it needs to be revised carefully. In addition, I would like to see more references on nanoparticles and their application, such as:

Baldelli, A., Ou, J., Barona, D., Li, W., & Amirfazli, A. (2021). Sprayable, superhydrophobic, electrically, and thermally conductive coating. Advanced Materials Interfaces8(2), 1902110.

Baldelli, A., Ou, J., Li, W., & Amirfazli, A. (2020). Spray-On Nanocomposite Coatings: Wettability and Conductivity. Langmuir36(39), 11393-11410.

Besides these details, the paper has a good flow and a clear purpose.

Author Response

Answer: We thank the reviewer for his/her positive feedback.

  • The revised manuscript has been proofread

Reviewer 2 Report

In this contribution by Lemine and co-workers, the authors used metal-based nanoparticles for heatings. The study is in line with the aim and scope of the journal. Unfortunately, the paper contains many shortcomings, which should be addressed before the work may be reconsidered for publication. Please refer to the suggestions given below:
1) Graphical abstract is not informative. 
2) The article cannot be reproduced as there is insufficient precision in the description of the experimental conditions. The description of synthesis is of a particularly low level of detail. Consequently, others cannot verify these findings and build on them, thereby making the article of low potential impact. 
3) It is not possible to ensure that the presented results are statistically significant if the authors did not conduct a proper error analysis. This is a must. 
4) Insets in the Figures are not visible. 
5) Adequate characterization (by SEM or TEM) should be carried out to show how the microstructure of the material is affected by the temperature treatment. 
3) Multiple "Time (Secondes)" should be corrected

Author Response

Reviewer #2: Comments and Suggestions for Authors

In this contribution by Lemine and co-workers, the authors used metal-based nanoparticles for heatings. The study is in line with the aim and scope of the journal. Unfortunately, the paper contains many shortcomings, which should be addressed before the work may be reconsidered for publication. Please refer to the suggestions given below:

1) Graphical abstract is not informative.

Answer:

We do our best to reflect the subject of the paper in the abstract; I think it is not mandatory for submission

2) The article cannot be reproduced as there is insufficient precision in the description of the experimental conditions. The description of synthesis is of a particularly low level of detail. Consequently, others cannot verify these findings and build on them, thereby making the article of low potential impact.

Answer:

We do agree with you, in the revised version we have added the following paragraphs. All changes are highlighted in blue (yellow color) in the text.

" In the first step, γ-Fe2O3 NPs were synthesized by a modified sol–gel process in supercritical conditions of ethyl alcohol (EtOH) following similar procedure as in our previous work [25]. 5g of iron (III) acetylacetonate [C15H12FeO6] was dissolved in a 30 ml of methanol under magnetic stirring for 15 min. The solution was then placed in an autoclave and dried under supercritical conditions of EtOH. The supercritical conditions of ethanol are Pc = 63.3 bar and Tc = 243 oC. The control of the heating of the autoclave was realised by temperature programmer.

             In the second step, TiO2 NPs enriched by Fe2O3 particles were obtained by dissolving of 6.72 mmol of titanium (IV)-isoproxide (Ti(iOPr)4, 97%) in a mixture of methanol and acetic acid (2 ml/ 2 ml). After 15 min of magnetic stirring, 50 mg of Fe2O3 NPs prepared in the first step was added to the solution and introduced in ultrasonic bath for 10 min. The solution was then introduced in an autoclave, and dried again in the supercritical condition of EtOH.

 "

3) It is not possible to ensure that the presented results are statistically significant if the authors did not conduct a proper error analysis. This is a must.

Answer:

We do agree with you and the errors are added to SAR values by using standard deviation. It is important to highlight that we did each experience twice and from that we deduced the error.

4) Insets in the Figures are not visible.

Answer:

It's done

5) Adequate characterization (by SEM or TEM) should be carried out to show how the microstructure of the material is affected by the temperature treatment.

Answer:

We did TEM to have an idea about the size but there is no temperature treatment after the synthesis except the temperature rise under AC magnetic field.

3) Multiple "Time (Secondes)" should be corrected

Answer:

It's done

Reviewer 3 Report

The aim of this paper is to prepare new materials based on maghemite resp. maghemite-TiO2 nanoparticles and assess their suitability for magnetic hyperthermia treatment.

The article is well written in language terms, both the English used and the way the authors structured the article. The experimental part lacks detailed procedures, which is especially important for publications in the materials field in order to obtain reproducible results. The scientific impact of the results obtained is moderate, it could in my opinion be improved a lot if experiments of simultaneous magnetic and photodynamic heating were conducted, as this is the whole idea about covering the nanoparticles with TiO2 in the first place. As I said, it would have been really great, but I think it is not necessary in order for the article to be accepted (I would still encourage the authors to think about it though). Based on this I would recommend the article to be accepted after major revisions.

A few detailed comments follow below:

- The English used in the article is very good, the red line in the article is easily visible; there are some few language mistakes such as missing articles or double spaces which can be corrected by the editing team once the article gets accepted I guess, I did not point out all of them

- The introduction part is well written, with enough citations as well as covering different aspects of the scientific subjects covered in the article

-The experimental section needs to be thoroughly updated: Even the experimental part in the cited ref 25 is not very descriptive, and even if it was a brief summary of the procedure would have been necessary in this article. As it is, the authors should provide a detailed procedure, so that somebody else who was not involved in this article can reproduce the synthesis and obtain the same material; especially in materials science even small differences in the procedure count; I am not an expert in supercritical drying, hence I cannot yet outline exactly what is necessary in this case, but at least origin and grade of used chemicals, exact quantity (g and moles) of the chemicals and solvents used, mixing conditions (t,T, (even more like stirring speed unless a real solution is obtained)), the names and producing companies of the devices used for supercritical drying (oven and autoclave? (the autoclaves which I know do not have an option that allows the taking out of the solvent, as would need to be done for supercritical drying)), yield of obtained product (in grams and possibly %), and the same for the coating with titanium dioxide; ideally an image or photo of the setup of supercritical drying (in supporting information?); the yield is especially useful since the authors claim that it is a good method to obtain large quantities of material without too much effort

-I could not see reference 27 cited in the article

-2.2 structural measurements: the formula is named after Paul Scherrer, with 2 r; the cited reference does not explain anything more about the equation, since it is just a self citation I would suggest to remove it (especially since it cited afterwards again) and (if the authors want to cite anything) replace it with either a review about the equation or an original article such as “Scherrer, P., Bestimmung der Größe und der inneren Struktur von Kolloidteilchen mittels Röntgenstrahlen. Nachr. Ges. Wiss. Goettingen, Math.-Phys. Kl. 1918, 2, 98-100“

-2.3 magnetic characterization: …magnetic characterization was performed…; ..at a magnetic field of…

-figure 1 and its explanation: I do not think from the XRD results you can conclude that it is an inverse spinel structure, only that it is a spinel structure; since you write the TiO2 peaks are both from anatase and rutile, please mark them differently in 1b; according to 1b you do have a significant phase transition between the 2 samples, in that hematite was formed, this should be mentioned in the discussion part(maybe even the amount of hematite in % calculated); just from XRD results one also cannot exclude the formation of magnetite; this is later on mentioned in the disussion of Mößbauer spectra but could also be mentioned here; since two different phases of TiO2 are formed, the crystallite size of both should be calculated; if possible also the size of hematite particles

-the IR analysis should also discuss what the organic compounds are most likely to be, since a C-O band indicates there are definitely organic contributions as well (together with the not mentioned alkyl stretching bands at <3000 cm-1)

-squid measurements: ref 25 can be cut from here, as it was stated above that the particles synthesized there are the same as here; the nonmagnetic nature of TiO2 and the lower volume fraction of maghemite are basically the same effect, maybe one can be cut out or the sentence altered in another way to demonstrate that

-3.2.2 end of 1st paragraph: …decrease of the saturation magnetization of Fe2O3-TiO2.

-is the hematite contribution recognizable in the Mössbauer spectra or is the percentage too small to be recognized?

-3.3.3, 2nd paragraph: …field amplitude….

-3.3.3, last line: ..commercial ferrofluids…

-It is great that the authors compared SAR values with those obtained from materials described in the literature; however since the authors themselves state that it is better to compare ILP values, I would like it if a similar comparison is done for ILP values of magnetic materials described in the literature (and not just commercially available ferrofluids). Since maybe there are not as many ILP values reported for maghemite-TiO2 particles, maybe other particles specifically designed for magnetic hyperthermia treatment (such as Fe3O4 based ones) could also be listed to have a table with 5-10 values for different materials.

-fig 6,7,9: in the figure the time unit should be written as seconds (without e); fig. 8 it is field amplitude

-For each of the magnetic hyperthermia experiments all details should be given. This includes for example the frequency, the field amplitude and solvent for 3.3.1, and both field amplitude and concentration for 3.3.4. Again, please make sure that somebody who is not one of the authors can reproduce the results if they want to with the information given in the article.

-conclusion: the high crystallinity of the samples are mentioned multiple times, but there is no proof in the discussion part mentioned that the samples have high crystallinity

Author Response

Reviewer #3: Comments and Suggestions for Authors

The aim of this paper is to prepare new materials based on maghemite resp. maghemite-TiO2 nanoparticles and assess their suitability for magnetic hyperthermia treatment.

The article is well written in language terms, both the English used and the way the authors structured the article. The experimental part lacks detailed procedures, which is especially important for publications in the materials field in order to obtain reproducible results. The scientific impact of the results obtained is moderate, it could in my opinion be improved a lot if experiments of simultaneous magnetic and photodynamic heating were conducted, as this is the whole idea about covering the nanoparticles with TiO2 in the first place. As I said, it would have been really great, but I think it is not necessary in order for the article to be accepted (I would still encourage the authors to think about it though). Based on this I would recommend the article to be accepted after major revisions.

Answer:

We thank the reviewer for his/her positive feedback and we do agree with him about the investigation of photothermal for magnetic hyperthermia, which is one of the objectives of this work. Our next step is the study of heating efficiency of the nanoparticles under an Ac magnetic field and laser source. It is expected that TiO2 dissipated heat when laser is applied, which can be combine with heat dissipated by maghemite under magnetic field.

A few detailed comments follow below:

- The English used in the article is very good, the red line in the article is easily visible; there are some few language mistakes such as missing articles or double spaces which can be corrected by the editing team once the article gets accepted I guess, I did not point out all of them

- The introduction part is well written, with enough citations as well as covering different aspects of the scientific subjects covered in the article

-The experimental section needs to be thoroughly updated: Even the experimental part in the cited ref 25 is not very descriptive, and even if it was a brief summary of the procedure would have been necessary in this article. As it is, the authors should provide a detailed procedure, so that somebody else who was not involved in this article can reproduce the synthesis and obtain the same material; especially in materials science even small differences in the procedure count; I am not an expert in supercritical drying, hence I cannot yet outline exactly what is necessary in this case, but at least origin and grade of used chemicals, exact quantity (g and moles) of the chemicals and solvents used, mixing conditions (t,T, (even more like stirring speed unless a real solution is obtained)), the names and producing companies of the devices used for supercritical drying (oven and autoclave? (the autoclaves which I know do not have an option that allows the taking out of the solvent, as would need to be done for supercritical drying)), yield of obtained product (in grams and possibly %), and the same for the coating with titanium dioxide; ideally an image or photo of the setup of supercritical drying (in supporting information?); the yield is especially useful since the authors claim that it is a good method to obtain large quantities of material without too much effort

Answer:

We do agree with you, in the revised version we have added the following paragraphs. All changes are highlighted in blue (yellow color) in the text.

" In the first step, γ-Fe2O3 NPs were synthesized by a modified sol–gel process in supercritical conditions of ethyl alcohol (EtOH) following similar procedure as in our previous work [25]. 5g of iron (III) acetylacetonate [C15H12FeO6] was dissolved in a 30 ml of methanol under magnetic stirring for 15 min. The solution was then placed in an autoclave and dried under supercritical conditions of EtOH. The supercritical conditions of ethanol are Pc = 63.3 bar and Tc = 243 oC. The control of the heating of the autoclave was realised by temperature programmer.

             In the second step, TiO2 NPs enriched by Fe2O3 particles were obtained by dissolving of 6.72 mmol of titanium (IV)-isoproxide (Ti(iOPr)4, 97%) in a mixture of methanol and acetic acid (2 ml/ 2 ml). After 15 min of magnetic stirring, 50 mg of Fe2O3 NPs prepared in the first step was added to the solution and introduced in ultrasonic bath for 10 min. The solution was then introduced in an autoclave, and dried again in the supercritical condition of EtOH.

"

-I could not see reference 27 cited in the article

Answer:

It was cited at the end of this sentence: …. Previous studies reported that SAR values could be influenced by different parameters, among them, size, structure, magnetic properties, preparation methods, concentration, and the frequency and amplitudes of the applied magnetic field [23-28].

Now in the text after removing self-citation, ref [ 28] becomes Scherrer et al…..

-2.2 structural measurements: the formula is named after Paul Scherrer, with 2 r; the cited reference does not explain anything more about the equation, since it is just a self citation I would suggest to remove it (especially since it cited afterwards again) and (if the authors want to cite anything) replace it with either a review about the equation or an original article such as “Scherrer, P., Bestimmung der Größe und der inneren Struktur von Kolloidteilchen mittels Röntgenstrahlen. Nachr. Ges. Wiss. Goettingen, Math.-Phys. Kl. 1918, 2, 98-100“

Answer:

We do agree with you.

The self-citation is removed and the original article is added as suggested by the referee.

-2.3 magnetic characterization: …magnetic characterization was performed…; ..at a magnetic field of…

-figure 1 and its explanation: I do not think from the XRD results you can conclude that it is an inverse spinel structure, only that it is a spinel structure; since you write the TiO2 peaks are both from anatase and rutile, please mark them differently in 1b; according to 1b you do have a significant phase transition between the 2 samples, in that hematite was formed, this should be mentioned in the discussion part(maybe even the amount of hematite in % calculated); just from XRD results one also cannot exclude the formation of magnetite; this is later on mentioned in the disussion of Mößbauer spectra but could also be mentioned here; since two different phases of TiO2 are formed, the crystallite size of both should be calculated; if possible also the size of hematite particles

Answer:

We do agree with you about the spinel phase and it is corrected in the text.

We did not calculate the crystallite for TiO2; we have mainly focused our attention on the maghemite phase.

-the IR analysis should also discuss what the organic compounds are most likely to be, since a C-O band indicates there are definitely organic contributions as well (together with the not mentioned alkyl stretching bands at <3000 cm-1)

Answer:

Our objective was only to show the presence of TiO2 as we focused on the smagnetism and heating

-squid measurements: ref 25 can be cut from here, as it was stated above that the particles synthesized there are the same as here; the nonmagnetic nature of TiO2 and the lower volume fraction of maghemite are basically the same effect, maybe one can be cut out or the sentence altered in another way to demonstrate that

Answer:

We do agree with you

[25] is removed and also this sentence: and also to the lower volume fraction of maghemite

-3.2.2 end of 1st paragraph: …decrease of the saturation magnetization of Fe2O3-TiO2.

-is the hematite contribution recognizable in the Mössbauer spectra or is the percentage too small to be recognized?

Answer:

The best fitting was found by neglecting hematite

-3.3.3, 2nd paragraph: …field amplitude….

Answer:

It's done

-3.3.3, last line: ..commercial ferrofluids…

Answer:

It's done

-It is great that the authors compared SAR values with those obtained from materials described in the literature; however since the authors themselves state that it is better to compare ILP values, I would like it if a similar comparison is done for ILP values of magnetic materials described in the literature (and not just commercially available ferrofluids). Since maybe there are not as many ILP values reported for maghemite-TiO2 particles, maybe other particles specifically designed for magnetic hyperthermia treatment (such as Fe3O4 based ones) could also be listed to have a table with 5-10 values for different materials.

Answer:

We do agree with the reviewer regarding ILP values but unfortunately many papers do not report ILP and calculated only SAR.

-fig 6,7,9: in the figure the time unit should be written as seconds (without e); fig. 8 it is field amplitude

Answer:

It's done

-For each of the magnetic hyperthermia experiments all details should be given. This includes for example the frequency, the field amplitude and solvent for 3.3.1, and both field amplitude and concentration for 3.3.4. Again, please make sure that somebody who is not one of the authors can reproduce the results if they want to with the information given in the article.

Answer:

We do agree with you

Information are added in the text and also in each figure.

-conclusion: the high crystallinity of the samples are mentioned multiple times, but there is no proof in the discussion part mentioned that the samples have high crystallinity

Answer:

We believe that XRD and TEM show that that the synthesized nanoparticles are with high crystallinity

Round 2

Reviewer 2 Report

The article may be accepted for publication.

For the future, it is advisable to measure more than 2 samples to do the statistics as n=2 seriously affects the confidence interval. Already at n=3, the situation becomes much better. 

Author Response

Dear Referee

Thanks for your advise. We do agree with you about the number of the samples in our coming papers

Reviewer 3 Report

-Experimental part: I would like to thank the authors for significantly updating the experimental part. However, as I wrote in my previous review, I still think some information should be added, such as origin (and quality in some cases) of the chemicals used (like: “Iron acetylacetonate (XX%) was obtained from YYY, ethanol (ZZ%) and acetic acid (AA%) from BBB, ..”), models of autoclave and ovens used (I still cannot imagine the way an autoclave for supercritical drying is built, hence I for one could not reproduce this synthesis if I wanted to), temperature programming (X°C/min till Y°C, hold for Z h, …..), and yield (in g, because it is probably difficult to calculate an exact percentage unless the iron content was measured). In the cited ref. 25 I also saw that particle size depends on stirring time: does this mean stirring time before the supercritical drying, or during? In this case I would also like to ask the stirring speed and temperature (I suppose room temperature?). Please do the same for both reaction steps.

-ref. 28: I am sorry if I was misunderstood in this case: What I meant by my suggestion was to remove the citation at the point where the Scherrer equation was mentioned in the article, not to remove the whole reference. Since it is cited (I think) 6 times in the original manuscript, and the authors only changed two of those, in the later parts of the article the original article by Scherrer is cited in a context where it does not make sense. The authors could of course remove those 4 citations too, but if they follow my original intention (which I guess was a bit misleadingly written) they can put the original reference 28 back in again, cite it once in the introduction and 4 times later on in the article, and only leave the citation of the Scherrer article (as ref 29) in the place where they use and explain the actual Scherrer equation; also you still did not correct his name (Scherrer not Scherer, at the end of page 6)

-Results-XRD: I agree that the maghemite is probably more important, but since you have the spectra and calculated the crystallite size for one modification of titanium oxide already, and since you did notice there are two modifications for titanium oxide in the sample, you should also distinguish between both of them and calculate the crystallite size for both. You also did not mark the peaks for rutile and anatase differently in Fig 1b. I also want to repeat that in my opinion the existence of hematite in the second sample IS a significant structural change, as it should influence the SAR values, among other things, and the amount of hematite as % of total iron oxides should be calculated (maybe also the size of hematite crystallites should be calculated if possible).

-Results-IR: Since the authors noticed and described the C-O band, even if their main intention is to discuss the formation of titanium oxide, the origin the C-O band should be discussed. And since the origin is most likely organic, the C-H stretching vibrations should also be listed and discussed in that connection.

-ILP value comparison: Probably fewer articles mention ILP than SAR, but it is still possible to find articles. Ten minutes of googling found me 3 articles which do have ILP values mentioned, two of them even for maghemite nanomaterials (I did not read the articles in detail, but checked that they were about iron oxide nanomaterials and had ILP values reported). The authors do not have to use these in their table, but I am sure they can find 10 articles on their own easily enough (maghemite, magnetite or ferrites since they had also ferrites listed in their SAR table, whichever they find)

M.C. Horny, J. Gamby, V. Dupuis, J.M. Siaugue, Magnetic Hyperthermia on gamma-Fe2O3@SiO2 Core-Shell Nanoparticles for mi-RNA 122 Detection, Nanomaterials-Basel, 2021, 11.

B.B. Lahiri, T. Muthukumaran, J. Philip, Magnetic hyperthermia in phosphate coated iron oxide nanofluids, J Magn Magn Mater, 2016, 407, 101-113.

  1. Bender, J. Fock, C. Frandsen, M.F. Hansen, C. Balceris, F. Ludwig, O. Posth, E. Wetterskog, L.K. Bogart, P. Southern, W. Szczerba, L.J. Zeng, K. Witte, C. Gruttner, F. Westphal, D. Honecker, D. Gonzalez-Alonso, L.F. Barquin, C. Johansson, Relating Magnetic Properties and High Hyperthermia Performance of Iron Oxide Nanoflowers, J Phys Chem C, 2018, 122, 3068-3077.

-mention of crystallinity in the conclusion: I talked to an expert on XRD and he said it is possible to conclude that there is no sign of an amorphous phase, although ideally the XRD pattern should have been extended to lower values of 2Theta to be really sure. (this is if high crystallinity is meant as opposed to being amorphous..if it was meant as large crystallite size it is obviously wrong, as the authors themselves calculated the crystallite size to be small) I will therefore accept the statement of high crystallinity in the conclusion, but the authors should still add a statement in the results/XRD part how they concluded that the sample has a high crystallinity. As for using TEM to conclude the high crystallinity I disagree. Firstly TEM only shows small parts of the sample, and secondly I had myself some samples that were amorphous according to XRD, but still showed very well formed shapes in TEM (in my case they were octahedra)

Author Response

Dear Reviewer

On the behalf of my colleagues, I would like to thank you your precious time in reviewing our paper and for their fruitful comments which will certainly strengthen our paper, we really appreciate very much. All comments are answered very carefully hereafter in blue and incorporated in yellow in the revised version of the manuscript.

-Experimental part: I would like to thank the authors for significantly updating the experimental part. However, as I wrote in my previous review, I still think some information should be added, such as origin (and quality in some cases) of the chemicals used (like: “Iron acetylacetonate (XX%) was obtained from YYY, ethanol (ZZ%) and acetic acid (AA%) from BBB, ..”), models of autoclave and ovens used (I still cannot imagine the way an autoclave for supercritical drying is built, hence I for one could not reproduce this synthesis if I wanted to), temperature programming (X°C/min till Y°C, hold for Z h, …..), and yield (in g, because it is probably difficult to calculate an exact percentage unless the iron content was measured). In the cited ref. 25 I also saw that particle size depends on stirring time: does this mean stirring time before the supercritical drying, or during? In this case I would also like to ask the stirring speed and temperature (I suppose room temperature?). Please do the same for both reaction steps.

-ref. 28: I am sorry if I was misunderstood in this case: What I meant by my suggestion was to remove the citation at the point where the Scherrer equation was mentioned in the article, not to remove the whole reference. Since it is cited (I think) 6 times in the original manuscript, and the authors only changed two of those, in the later parts of the article the original article by Scherrer is cited in a context where it does not make sense. The authors could of course remove those 4 citations too, but if they follow my original intention (which I guess was a bit misleadingly written) they can put the original reference 28 back in again, cite it once in the introduction and 4 times later on in the article, and only leave the citation of the Scherrer article (as ref 29) in the place where they use and explain the actual Scherrer equation; also you still did not correct his name (Scherrer not Scherer, at the end of page 6)

Answer:

  • We do agree with you, We misunderstood what you mean in the first revision.

Reference 28 in first dratf is added now as ref [22] in the introduction; after removing another reference which is not really significant in the introduction.

  • It is cited correctly in the text.
  • Scherrer not Scherer: done

-Results-XRD: I agree that the maghemite is probably more important, but since you have the spectra and calculated the crystallite size for one modification of titanium oxide already, and since you did notice there are two modifications for titanium oxide in the sample, you should also distinguish between both of them and calculate the crystallite size for both. You also did not mark the peaks for rutile and anatase differently in Fig 1b. I also want to repeat that in my opinion the existence of hematite in the second sample IS a significant structural change, as it should influence the SAR values, among other things, and the amount of hematite as % of total iron oxides should be calculated (maybe also the size of hematite crystallites should be calculated if possible).

Answer:

The two phases are added on Fig1.b and the size is deduced

-We do agree with you about the presence of hematite but unfortunately we cannot determine the amount of hematite from our current results. We have not the possibility to do EDAX or XPS in our laboratory.

  • Results-IR: Since the authors noticed and described the C-O band, even if their main intention is to discuss the formation of titanium oxide, the origin the C-O band should be discussed. And since the origin is most likely organic, the C-H stretching vibrations should also be listed and discussed in that connection.
  • Answer:
  • C-H stretching is added in the discussion

But as explained in the first version, our focus is on magnetic and hyperthermia properties.

  • ILP value comparison: Probably fewer articles mention ILP than SAR, but it is still possible to find articles. Ten minutes of googling found me 3 articles which do have ILP values mentioned, two of them even for maghemite nanomaterials (I did not read the articles in detail, but checked that they were about iron oxide nanomaterials and had ILP values reported). The authors do not have to use these in their table, but I am sure they can find 10 articles on their own easily enough (maghemite, magnetite or ferrites since they had also ferrites listed in their SAR table, whichever they find)

Answer:

ILP is added in the table

-mention of crystallinity in the conclusion: I talked to an expert on XRD and he said it is possible to conclude that there is no sign of an amorphous phase, although ideally the XRD pattern should have been extended to lower values of 2Theta to be really sure. (this is if high crystallinity is meant as opposed to being amorphous..if it was meant as large crystallite size it is obviously wrong, as the authors themselves calculated the crystallite size to be small) I will therefore accept the statement of high crystallinity in the conclusion, but the authors should still add a statement in the results/XRD part how they concluded that the sample has a high crystallinity. As for using TEM to conclude the high crystallinity I disagree. Firstly TEM only shows small parts of the sample, and secondly I had myself some samples that were amorphous according to XRD, but still showed very well formed shapes in TEM (in my case they were octahedra)

Answer:

High crytsallinity is removed from the conclusion

We believed that the most important that the synthesized nanoparticles have a small size and that will be benefits for future in-vitro hyperthermia measurements.